# Mapping the Tray of Electron Beam Melting of Ti-6Al-4V: Properties and Microstructure

**DOI:** 10.3390/ma12091470

**Published:** 2019-05-07

**Authors:** E. Tiferet, M. Ganor, D. Zolotaryov, A. Garkun, A. Hadjadj, M. Chonin, Y. Ganor, D. Noiman, I. Halevy, O. Tevet, O. Yeheskel

**Affiliations:** 1Materials Department, Nuclear Research Center Negev, P.O. Box 9001, Be’er Sheva 8419001, Israel; ganorm55@gmail.com (M.G.); sivan.moore82@gmail.com (A.H.); noimand@hotufi.net (D.N.); tevet.ofer@gmail.com (O.T.); 2AM Center, Rotem Ind. Mishor Yamin 86800, Israel; michaelc@rotemi.co.il (M.C.); Yarong@rotemi.co.il (Y.G.); 3Israel Institute of Metals, Technion Haifa 32000, Israel; denisz@trdf.technion.ac.il (D.Z.); agar@trdf.technion.ac.il (A.G.); 4Nuclear Engineering Unit, Ben Gurion University, Be’er Sheva 8410501, Israel; halevy.itzhak.dr@gmail.com; 5Physics Department, Nuclear Research Center Negev, P.O. Box 9001, Be’er Sheva 8419001, Israel; 6Materials Engineering Department, Ben Gurion University, Be’er Sheva 8410501, Israel

**Keywords:** Powder bed, fatigue, Hot Isostatic Pressure, Electron Beam Melting

## Abstract

Using an electron beam melting (EBM) printing machine (Arcam A2X, Sweden), a matrix of 225 samples (15 rows and 15 columns) of Ti-6Al-4V was produced. The density of the specimens across the tray in the as-built condition was approximately 99.9% of the theoretical density of the alloy, ρ_T_. Tensile strength, tensile elongation, and fatigue life were studied for the as-built samples. Location dependency of the mechanical properties along the build area was observed. Hot isostatic pressing (HIP) slightly increased the density to 99.99% of ρ_T_ but drastically improved the fatigue endurance and tensile elongation, probably due to the reduction in the size and the distribution of flaws. The microstructure of the as-built samples contained various defects (e.g., lack of fusion, porosity) that were not observed in the HIP-ed samples. HIP also reduced some of the location related variation in the mechanical properties values, observed in the as-printed condition.

## 1. Introduction

Additive manufacturing (AM) is a method of transforming digital design files into functional engineering products. Offering flexibility in design and environmental advantages, the technology also enables engineers to produce low volumes of unique objects in an economical way. The methods and materials may vary, but all AM processes typically work by building their components up one layer at a time. One of the most commonly used methods is powder bed fusion AM (PBF-AM), which involves focusing an energetic (laser or electron) beam in order to melt specific locations of a powder layer spread on a base plate. A second powder layer is then spread on top of the first one, and the fusion process is repeated [1]. The energy emitted is absorbed by powder particles via both bulk-coupling and powder-coupling mechanisms [2]. The resulting transient temperature field, characterized by high temperatures and rapid solidification rates, is formed concomitantly during the interaction between the beam and the powder bed and has a significant effect on defect formation, final microstructure, and mechanical properties of the components [3]. In addition, the transient thermal behavior is controlled by processing parameters such as material properties, beam characteristics, and scan speed and strategy. In laser powder bed fusion (L-PBF), due to the fact that there is relatively little preheating, the high cooling rates during melting and solidification cause internal residual stresses, which in turn induce plastic deformation within thin metal layers [4]. Sacrificial support structures do not always compensate for this local plastic deformation, leading to the formation of cracks and product or support structure failure. In fact, the complex metallurgical nature of L-PBF involves multiple modes of heat, mass, and momentum transfer that often lead to uncertainty concerning the final part mechanical properties [5,6]. Furthermore, post-processing heat treatments are inevitable in the case of L-PBF, whereas in electron beam melting (EBM), it is not mandatory to stress relief after AM builds. A common post process is hot isostatic pressing (HIP), which is used mainly to eliminate pores in castings [7,8]. Since pores and other defects affect the fatigue life of materials [9], HIP can improve the fatigue endurance limit of samples produced by both L-PBF and EBM [10,11,12]. Several materials, including steel, titanium, and aluminum alloys, manufactured by AM have been studied for defect formation as well as for mechanical performance [13,14]. The most well-studied titanium alloy applicable in the aerospace and biomedical fields is Ti-6wt%Al-4wt%V (Ti-6Al-4V); these studies have shown that the quasi-static tensile strength of AM manufactured Ti–6Al–4V is comparable to that of conventionally-manufactured alloys [3,9,15,16,17]. In addition, it has been shown that with conventional manufacturing, elongation of Ti-6Al-4V alloys gradually decreases as oxygen concentration increases [16,17]. In contrast, for AM using direct energy deposition (DED), an increase in oxygen concentration promotes the formation of Ti-α at grain boundaries, Ti-β inside grains, and finer Ti-α laths. Consistent experimental results have shown an increase in tensile properties with increased oxygen, with no significant effect on elongation [18]. When utilizing EBM, it can be assumed that objects can be placed at any location in the powder bed tray. 

### HIP as a Post-Process for Additively Manufactured End Products

Many engineering parts once produced from powders using conventional methods such as cold compaction followed by sintering are now produced using AM [19]. However, these products often contain material imperfections in the form of pores and/or lack of melting [3]. In certain applications, mostly those where cyclic loading is incorporated, these imperfections are detrimental to the material’s mechanical properties and thus must be eliminated. The HIP process consists of the concomitant application of high temperature and pressure on parts that are placed in a pressure vessel. The temperatures employed in metals are in the range of 0.5 Tm to 0.9 Tm, where Tm is the melting temperate [K] of the material, while the typical pressure utilized in most HIP processes of metallic objects is 100 MPa [7]. Applying high temperature and pressure on a material results in the initiation of creep mechanisms in addition to enhancing the bonding of adjacent surfaces by diffusion while also closing existing porosity [7,20]. AM offers intricate geometries that are designed for a specific demand. Residual flaws such as lack of fusion, cracks, or spherical porosity affect the fatigue life. In order to reduce such flaws population, post processing HIP is required to enhance the properties of products built via AM, especially those subjected to fatigue conditions [10]. Yet, the fact that the HIP process takes place at elevated temperatures may actually have adverse effects on the properties that it was originally intended to enhance. For example, the unique microstructure that is typical of materials built via AM may be completely altered as a consequence of the high temperatures employed. Recently, a Laser-PBF-AM study conducted on Inconel showed improved mechanical properties with an HIP post-process [21], as well as the partial elimination of inherently imposed pores. Since different locations in the build tray impose different thermal histories [22], it may be that defect variation is correlated with their position on the build plate. 

The objective of the current study is to present a map dependency of the physical and the mechanical properties and the microstructure of AM-EBM Ti-6Al-4V within the powder bed volume were it is located. Additional aspects of HIP influence on microstructure and mechanical properties are also presented and discussed.

## 2. Experimental

### 2.1. Research Methodology

The methodology used in this study was based on relationships between the raw material AM process parameters and the microstructure properties with and without post treatment, as seen in Figure 1. This was done by utilizing powder after multiple uses (see Section 2.2), single printing parameters (see Section 2.3), and a single post-treatment cycle (see Section 2.4), measuring microstructure mainly via SEM (see Section 2.5) and neutron diffraction, reported elsewhere [23]. Physical properties, density, elastic properties, tensile strength, and fatigue were correlated to the location on the tray. 

### 2.2. Powder Characteristics

The Ti-6Al4V powder was supplied by Arcam (batch 1410, in compliance with the American Society for Testing and Materials, ASTM F2924 standard), and used in the previous 38 cycles. Particle size distributions of the powder in cycle 39 (C-39) for D_10_, D_50_, and D_90_ values were 66, 84, and 120 μm, respectively. The sphericity ratio was 0.9 (where 1 represents a perfect sphere), and powder density was determined using pycnometry to be 4427 ± 15 kg/m^3^. As expected [24], reuse of powder did not affect particle size or shape distribution.

### 2.3. Printing Theme

The Ti-6Al-4V alloy samples were produced by an Arcam A2X machine (Arcam AB, Mölndal, Sweden) using recycled grade 5 powder. The acceleration voltage remained constant at 60 kV. Vacuum conditions were 10^−5^ mbar of the initial vacuum with a needle valve providing a constant 10^−3^ mbar helium environment.

Powder was spread on a platform of a pre-heated plate. Each powder layer was pre-heated in order to attain a temperature between 550 and 700 °C (~0.5 T_M_). The main reason for conducting the pre-heating stage was to improve the energy deposition efficiency. When an electron beam interacts with a cold powder, part of the energy is not deposited (and transferred to heat) as intended. Instead, some powder gets charged by the electrons [25], and a kinematic phenomenon of recoiled powder particles emerging from the powder bed occurs. The pre-heat ensures that the efficiency of the energy deposition is optimal.

The tray consisted of a matrix of 15 rows (labeled A to O) by 15 columns (labeled numerically). Some of these rows contained cuboid samples (10 mm by 12 mm by 70 mm height) intended for non-destructive evaluation of the build process quality. Most of the samples were rods with a diameter of ~10 mm that were used for testing mechanical properties (Figure 2). The height of all samples was ~70 mm, each one designated by a row and a column notation, e.g., A3 refers to the third sample (column) in row A. The intersection of lines A, H, and O with columns 2, 4, 6, 10, 12, and 14 marks specific samples that were tested.

The electron beam path and the parameters (e.g., beam speed and current) were guided by an algorithm (proprietary) formulated by Arcam. The purpose of this algorithm was to maintain a constant heat deposition (J/mm^2^) in each area in a single layer; additionally, all layers should have had a similar heat deposition. In the current work, the geometries were simple, therefore equal heat deposition was achievable. Nevertheless, since the melted area in each layer was relatively large (i.e., more than 50%), it was necessary to manage the melt order in an attempt to keep the heat distribution as even as possible. Various STL (stereo lithography) builds were tried, yet most of them resulted in the smoke [25] that evolves when an electron beam interacts with cold powder. For the successful build, 15 STLs were used for the various rows, and 1 STL was used for the rectangular rods.

### 2.4. HIP Post-Processing

Post-processing of AM specimens was carried out using a laboratory-sized HIP apparatus under high purity (99.99% pure) Ar gas at a pressure of 120 MPa at 920 °C for two hours in accordance with an appropriate standard [26]. The printed rods were wrapped with proprietary protective foil to reduce the likelihood of reaction with minute impurities of the gas. 

### 2.5. Characterization Methods

Powder PSSD (particle size and shape distribution) was evaluated using a Qicpic instrument (Sympatec, Germany), while microstructure and fracture surface were studied utilizing high resolution SEM. Chemical composition of the metallic elements was measured by EDS (energy dispersive spectroscopy); evolved gas was measured by Leco (™). The density of the built specimens was measured using the Archimedes method described previously [27,28], and gas pycnometery was utilized to measure powder density. Samples for tensile and fatigue testing were machined to a surface finish of Ra = 0.4 − 0.6 < 1 μm. Tensile testing was carried out according to ASTM F2924–14 [26,29] at 24 °C using an Instron 3369 testing machine (50 kN load cell) and an Instron 2620-602. Elongation was measured based on a 20 mm length (about four times the diameter of the specimen). The tensile test was carried out at a strain rate of 0.005 min^−1^ until 2% total elongation was achieved. Then, the crosshead speed was changed to 1 mm per minute. The fatigue test was performed according to the ASTM E466 Standard with a focus on the high cycle section. The force controlled constant amplitude uniaxial loading was carried out using an Instron 8801 testing machine (Dynacell, Dynamic Load Cell +/− 100 kN) under load control with a sinusoidal waveform. Specimens were fixed using Instron fatigue-rated mechanical wedge grips. The Young’s moduli of the as-built and the HIP samples were calculated based on OLS (ordinary least squares) regression as per Section 0.2–0.5 of the proof stress. The elastic moduli of limited samples were also measured according to the pulse echo method described previously [25,30]. For fatigue, cyclic loading was applied in air at 23 ± 2 °C with a load ratio of R = 0.1 and a frequency of 30 Hz through the end of the test. Most specimens were tested to failure, while for some samples following HIP, the tests were stopped (i.e., run out) at 3 × 10^6^ cycles, and a few were stopped after 1 × 10^7^ cycles had elapsed. 

## 3. Results

### 3.1. Bulk Density

The average density, $\bar{x},$ and the standard deviation, S, of more than 140 as-built samples were 4427.6 ± 1.2 kg/m^3^, while the average density and the standard deviation of more than 50 samples that underwent HIP were 4432.1 ± 1.45 kg/m^3^. The theoretical density (ρ_T_) of the material was calculated by adding three standard deviations, 3σ = 3 × 1.45/√50, to the average of the HIP-ed samples, resulting in ρ_T_ = 4432.7 kg/m^3^. The relative density, ρ*, was ρ* = ρ/ρ_T_, where ρ was the density. Figure 3 shows a map of the relative density of AB (as-built) samples for which the average was 0.9991 with no dependency in location and with a variation of less than 0.2%. After HIP treatment, ρ* improved to a value of 0.9998 of the full density with a variation of less than 0.035%. 

### 3.2. Chemical Composition

Results were obtained from specimens A3, H3, and O3; averages and standard deviation are shown in Table 1. Note that the oxygen content exceeded ASTM F2924 standard requirements. 

### 3.3. Microstructure 

The microstructure of the as-built Ti-6Al-4V samples is shown in Figure 4. As expected, an α-β mixture with a very fine lamellar morphology was observed, in which the β phase is the white phase between lamellae of α, the gray phase. In samples taken from the HIP-ed samples, the microstructure was still lamellar with coarser lamellae (Figure 4B). It was recently shown using neutron diffraction that the β content in samples from a similar EBM process was <1 wt%. [31] This β phase content was analyzed using the SEM micrographs (Figure 4A,B), and it was determined that for the as-built sample and for the HIP-ed samples, the contents were about 2.5 ± 1.4% and about 5.8 ± 1.5% wt%, respectively. Such an increase is consistent with the increase in beta phase content upon heating up to 1000 °C [26,29]. 

### 3.4. Tensile Properties and Fractography

Results of the tensile tests for the as-built samples presented in Table 2 show that the proof stress, or the plastic strain stress of 0.2% (R_p0.2_), and the ultimate tensile stress (R_m_) were higher by ~200 MPa than both ASTM F3001 and ASTM F2924 standard requirements. Low ductility in some of the samples that did not reach these required standards was observed. Variation in properties depending upon location was pronounced for the reduction in area and elongation, both with relatively high standard deviations, while the standard deviations of R_m_ and R_p0.2_ were much lower (Table 2). Upon correlating the properties with the printing order, further insights were revealed. Figure 5 shows the relationship between the engineering ultimate strength and the fracture strain; it should be noted that the middle line possesses a greater elongation compared to that in samples located closer to the circumference. Minor differences were observed when applying the ultrasound pulse echo method to measure the elastic moduli in small slices (~5 mm) taken from the bottom of selected rods (Table 3). 

Following HIP treatment, the specimens’ reduction of area and elongation improved significantly, while dependency of the location across the tray became insignificant (Figure 6). Hence, homogeneity of mechanical properties improved. In a single sample yielding the lowest value for reduction area (A4), a surface defect caused during the machining of the specimen was observed. Excluding this one sample, variation in the reduction of area was about 43%±3%, whereas for the AB samples, variation was ±11% (Table 2). Thus, it appears that HIP significantly improved average elongation and reduction of area, while at the same time slightly decreased the R_m_ and R_p0.2_ (about 5%), suggesting in this composition a Hall-Petch-like behavior. Additionally, Figure 7 presents typical tensile results of AB and HIP samples that failed prematurely. This early fracture in AB samples was initiated by three AM-related phenomena: lack of fusion, overheating, and porosity (Figure 8). Sample A2 showed large porosity and one lack of fusion defect (Figure 8A), sample A14 showed several lack of fusion defects and small size porosity (Figure 8B), while sample B8’s overheated area was evident (Figure 8C). No such defects appeared in the HIP sample fracture surface (Figure 8D). Finally, the difference in strain-to-fracture in samples A2 and A14 is discussed in Section 4.3.

### 3.5. Fatigue Tests and Fractography

The results of the fatigue tests for both as-built and HIP-ed samples (Figure 9) indicate that the fatigue limit (N_f_ = 10^7^ cycles) for the HIP-ed samples was about 550 MPa, and the estimated fatigue limit for the AB state was below 300 MPa. Similar to the tensile results, a pronounced improvement in fatigue properties after HIP was observed. Figure 10 shows the comparison in number of cycles between HIP and as-built samples when tested at a stress of 623 MPa. As demonstrated recently [10], voids were most often the source of fatigue crack initiation in EBM as-built samples; the improvement in fatigue strength could be attributed to the reduction in the number of initiation sites through the internal pore and the void closure during HIP. Figure 11 presents fracture surfaces of typical samples. It should be noted that remnants of printing defects were not observed on the HIP-ed samples. However, in the as-built samples, a crack was initiated by a fusion defect (overheating) in the bulk of the rod that, incidentally, was close in proximity to the machined fatigue sample.

## 4. Discussion

### 4.1. Thermal Management of Built Tray

The geometry of the specimens in the current study was simple, thus the average of the energy deposition between layers or between the areas in each layer was inherently imposed by the CAD (computer assisted design) model. Even though the heat depositions in different locations for a given layer were aimed at being similar, the thermal history of specimens created in the center of the tray was different from the samples originating at the edges of the tray. This variation in temperature was the main driving force for the variation in the mechanical properties. Nevertheless, since the melted area in each layer was relatively large (i.e., more than 50%), it was necessary to manage the melt order in an attempt to keep the heat distribution in every layer as even as possible. Upon splitting the build geometry to a single STL for each sample, the melt sequence became much more even (The reported tray was “printable” only when splitting the built file into eight individual STL (this work was done with the Arcam 4.2 version, and today, with 5.2 version, this is done automatically)). This even distribution was the key factor that prevented relatively cold areas on the built tray. Heat management prevented the electron beam from interacting with cold powder, and smoke generation was limited. Therefore, managing the heat during the build is important not only in order to maintain a constant melt pool size to achieve homogeneous microstructure and mechanical properties [9], but also to sustain production when generating large volume builds. With all the caution taken to achieve homogeneous printing, Figure 8C shows an example of an uncontrolled, overheated area.

### 4.2. Location Dependency of As-Built Samples

As described in Section 4.1, an attempt to maintain similar heat deposition along the built was done, yet AM-PB-EBM had a noticeable location dependency that dictated mechanical property variation across the build area (Figure 6). Front and rear built borders (rows O and A) were subjected to relatively fast cooling rates, which might explain why the inner part of the built tray possessed better physical properties. Differing thermal histories, a direct result of location, is the subject of ongoing research. 

### 4.3. HIP Influence 

As expected, the microstructure of the AB EBM Ti-6Al-4V samples consisted of an α-β mixture with a very fine lamellar morphology in which the β phase was located between the lamellae of α. When executing an HIP heat treatment, a sufficiently high temperature was necessary to eliminate pores and lack of fusion defects with minor changes in the unique AM microstructure. At 920 °C, the microstructure of the HIP samples was still lamellar, although the lamellae size coarsened, which led to a slight decrease in yield strength as well as in ultimate tensile strength [11]. Proof, or the yield stress (YS), and the ultimate tensile strength (UTS) of the as-built samples were measured to be more than 200 MPa, greater than required by both ASTM F3001 and ASTM F2924 standards (Table 2). Ductility of the as-built samples was low, ~10% on average, and in five of the 12 samples, it was lower than standards dictate. The YS and the UTS of the HIP-ed samples were 7% lower compared to the AB samples. When excluding the result of the one sample that failed due to a machining defect, the averages of YS and UTS were 976 MPa and 1090 MPa, respectively, or about 20% higher than standards require. In addition, remarkable ductility improvement of HIP samples was observed, from an average of ~10% in the AB samples to ~20% in the HIP-ed samples. The main reason for this increase was the reduction in size and population of defects of the as-built samples. Defects such as porosity, lack of fusion, and overheating were successfully healed by the HIP process. Moreover, fatigue limit for HIP-ed samples was much higher with respect to that in the as-built samples, above 550 MPa for the HIP-ed samples versus around or even below 300 MPa for the as-built ones (Figure 11). The fatigue endurance limit (N_f_ = 10^7^ cycles) for the as-built samples was found to be~250 MPa [10]; other studies have previously shown that the stress for this limit is much higher after HIP [10,32]. Whereas the fatigue crack initiation of the HIP samples started inside the bulk, for as-built samples, it started from the printing defect located below the sample surface. A similar observation held true for tensile tests. A few examples are related herein, e.g., in the fracture of sample A2, large porosity and one large lack of fusion defect were detected (Figure 9A). This sample failed with a strain of ~13%, while sample A14, which contained two large lack of fusion defects and small porosity (Figure 8B), failed at 6%. The reason for this inconsistency may be related to the critical flaw size of the two combined lack of fusion defects in A14 as compared to the combined lack of fusion and large porosity in A2. Further discussion of this hypothesis is irrelevant here due to a scarcity of data. However, supporting evidence on the importance of lack of fusion as a source of fracture of PBF-AM also existed for sample N4 (Figure 11C), which prematurely failed in fatigue after ~10^5^ cycles at a stress of 415 MPa (Figure 9). It is suggested here that HIP heals not only porosity—a rounded defect—but also planar defects such as lack of fusion or overheating. Thus, HIP reduces the population of internal defects below a certain critical level. This reduction leads to dramatic improvement in elongation and fatigue cycles at any applied level of stress. Furthermore, HIP demonstrates a dramatic reduction in variation of mechanical properties depending upon location on the tray. These values were achieved in the presence of relatively high (0.34%) oxygen content. In commercial Ti64 [15], elongation was reduced linearly from 9% in 0.125% oxygen content to ~7% for 0.35% oxygen content. The high elongation and fatigue results of PBF-EBM in the current study were likely an outcome of the fine microstructure, which compensated for the deleterious effects of the elevated presence of oxygen in pure Ti and commercial Ti64 [15].

## 5. Summary

In this study, it was demonstrated that a very high areal density (>50%) was successfully printed using PBF electron beam melting with an average sample density of ~0.999 of the theoretical value. Mapping the mechanical properties across the printed tray revealed that, while density variation across samples was very small (<0.5%) and variation of tensile strength was small (~2%), the tensile elongation of the as-built samples showed large scatter (~40%) and some dependency on the location across the tray. The fatigue properties of the as-built samples were found to be low, with an estimated fatigue stress at 10^7^ cycles below 300 MPa. Fractography of the samples following tensile and fatigue testing showed that planar defects such as lack of fusion and overheating were the major defects causing premature failure. However, while performing HIP at 920 °C only slightly increased density to 0.9999 of the theoretical density, it drastically improved elongation by ~200% with only a slight decrease (~5%) in proof stress and ultimate stress. In addition, the variation in properties relative to the location on the tray diminished, with the strength and the elongation of HIP-ed samples far surpassing the requirements dictated by ASTM F2924. Finally, and most importantly, was the improvement in fatigue life, whereby the fatigue stress for failure after 10^7^ cycles was above 550 MPa. Our main claim here, therefore, is that HIP not only reduces the size and the distribution of flaws, but also significantly improves elongation and fatigue limit. Mechanical property variation decreased significantly as well, and no dependency in location could be noted after HIP treatment.

## Figures and Tables

**Figure 1 materials-12-01470-f001:**
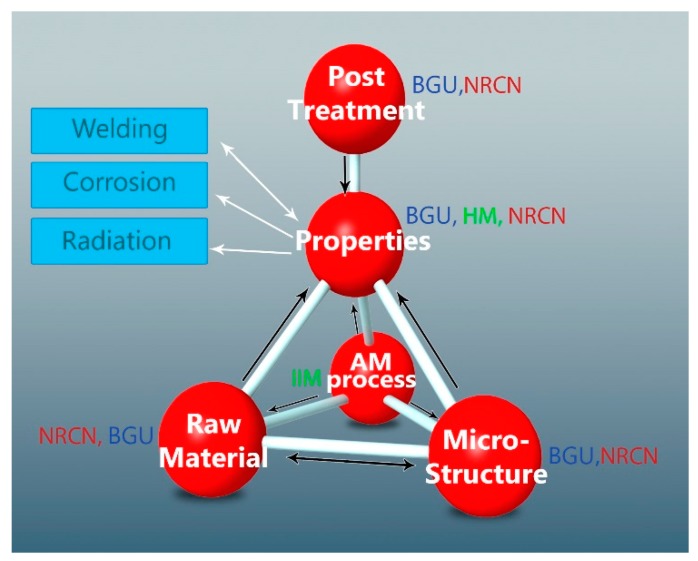
Material-processing properties and microstructure relationships examined.

**Figure 2 materials-12-01470-f002:**
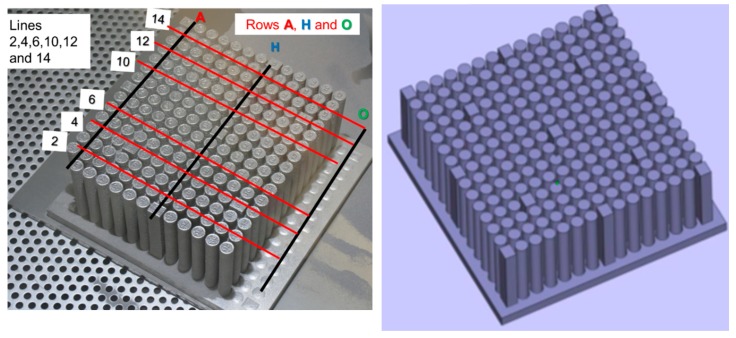
Samples and coordinates designating their location (left) and the computer-assisted design (CAD) file used for the electron beam melting (EBM) build (right).

**Figure 3 materials-12-01470-f003:**
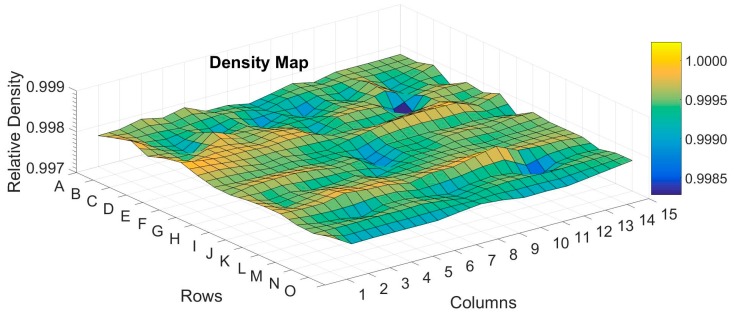
Mapping the variation of relative density across the built tray.

**Figure 4 materials-12-01470-f004:**
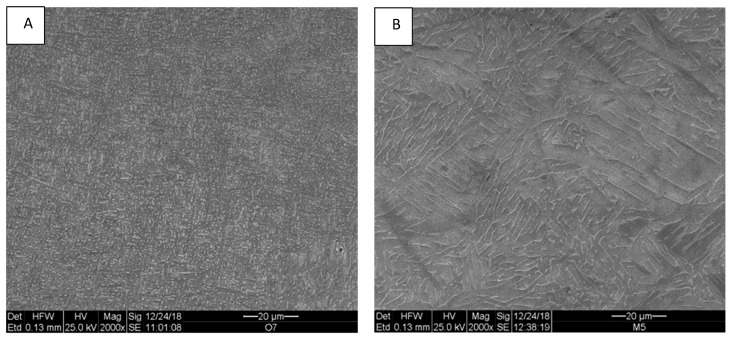
Microstructure of as-built (**A**) and hot isostatic pressing (HIP)-ed (**B**) samples.

**Figure 5 materials-12-01470-f005:**
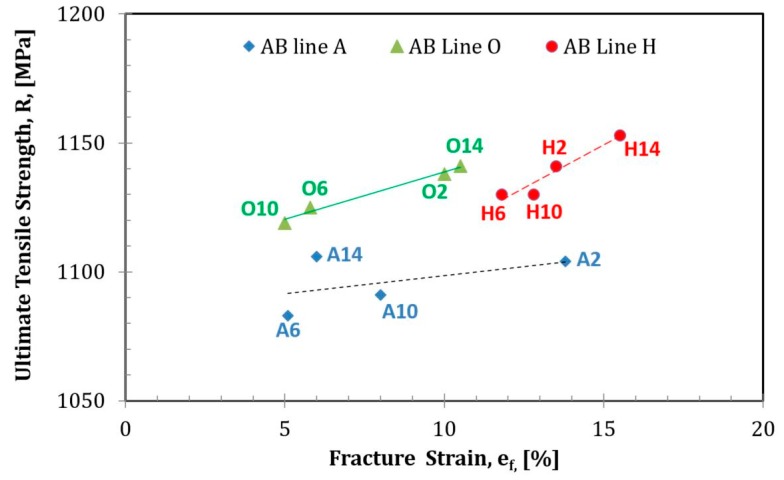
Change in engineering ultimate tensile strength with engineering fracture strain in as-built (AB) samples as a function of printing location in rows A, H, and O.

**Figure 6 materials-12-01470-f006:**
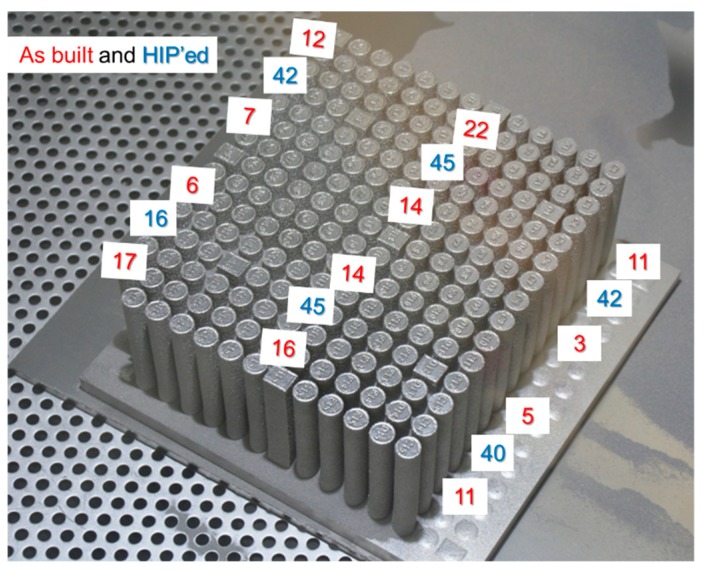
The reduction of area (%) as a function of location of the samples on the tray. The numbers represent the reduction of area in the as-built state (red) and after HIP (blue).

**Figure 7 materials-12-01470-f007:**
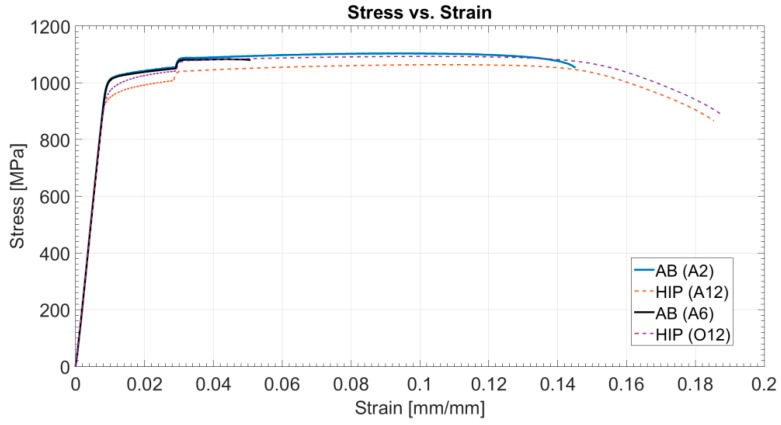
Typical engineering stress–strain curves for AB and HIP samples.

**Figure 8 materials-12-01470-f008:**
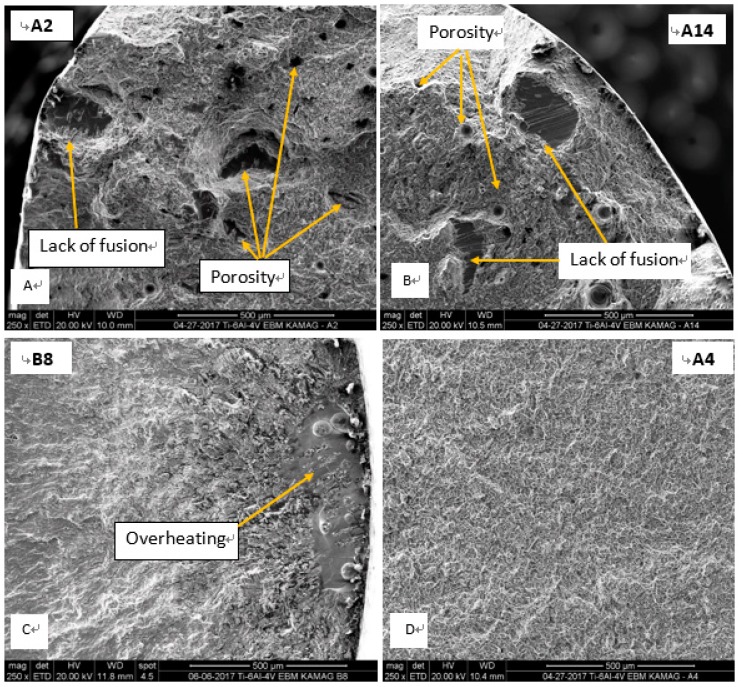
Fractography of AB samples depicting large porosity (**A**), large lack of fusion (**B**), large overheating (**C**), and of HIP-ed sample (**D**), A4, showing a much sounder microstructure than the AB.

**Figure 9 materials-12-01470-f009:**
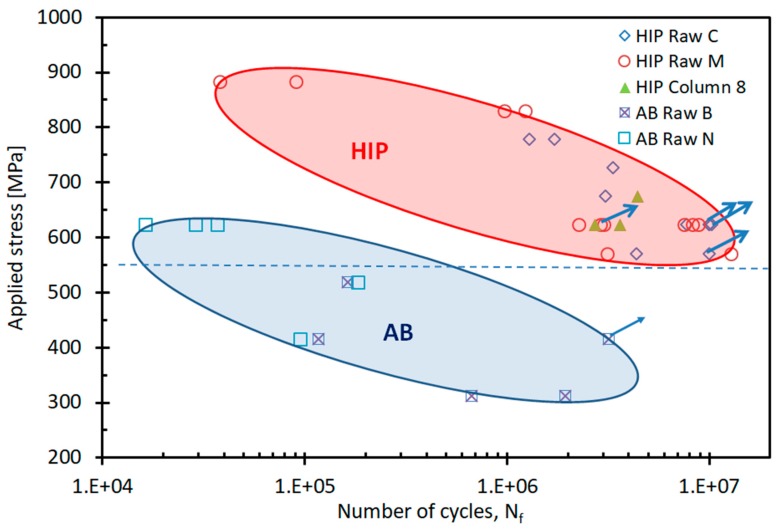
Fatigue tests on AB samples and HIP. Arrows indicate deliberate run outs.

**Figure 10 materials-12-01470-f010:**
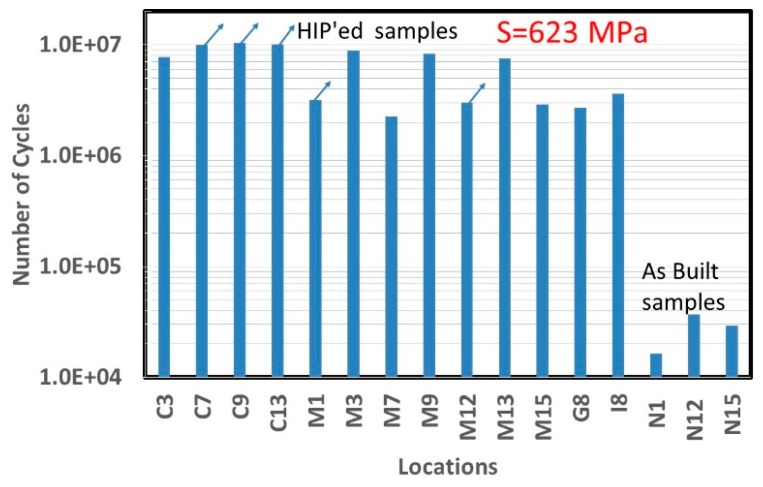
Number of cycles to failure at stress of 623 MPa in HIP-ed samples from row C and row M and a few samples from column 8; AB samples are from row N. Arrows indicate “run outs” at either 3 × 10^6^ or 1 × 10^7^ cycle.

**Figure 11 materials-12-01470-f011:**
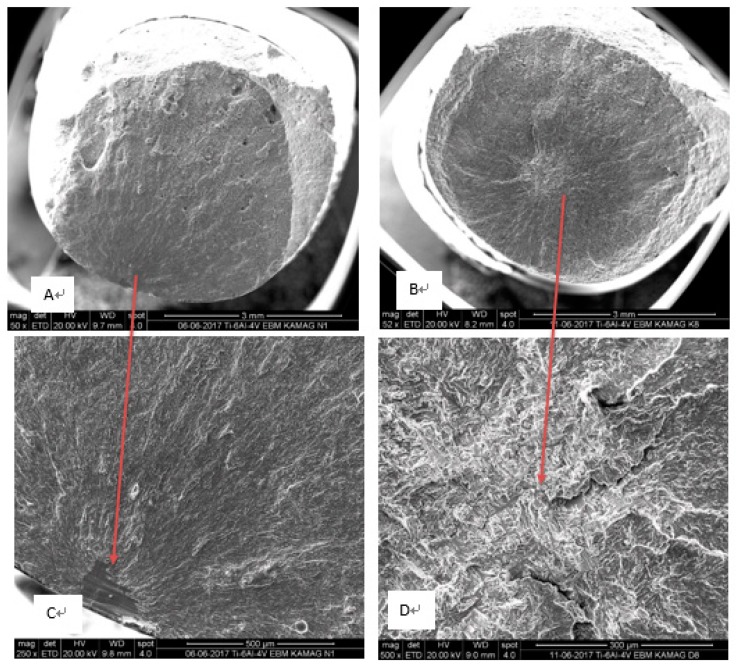
Macrographs of the fracture surface of the fatigue samples as-built (**A**) and HIP-ed (**B**). Fractography reveals a lack of fusion defect in the as-built sample as a source of early fracture (**C**), while the fracture of the HIP sample was initiated in the bulk (**D**).

**Table 1 materials-12-01470-t001:** Averages and standard deviations, S^†^, of elemental composition of the sample across the tray compared to the requirements of a proper ASTM standard.

Element	Ti	Al	V	Fe	O	C	H
ASTM F2924	Balance	5.5–6.75	3.5–4.5	<0.3	<0.2	<0.08	<0.0015
Average ±S ^†^ [%]	89.0 ± 0.22	6.5 ± 0.13	3.86 ± 0.04	0.27 ± 0.05	0.34 ± 0.01	0.031 ± 0.001	0.0027 ± 0.0008

† $S=\sqrt{\frac{\left(x^{2}-{\bar{x}}^{2}\right)}{n-1}}$ where $\bar{x}$ is the average, n is the number of measurements, and x is the result.

**Table 2 materials-12-01470-t002:** Averages and standard deviations, S, of engineering tensile test results: Young’s modulus, E, proof strength, R_p0.2_, ultimate tensile strength, R_m_, elongation, e, and reduction of area, A, for the as-built and the HIP samples. The results for HIP samples, excluding sample A4, are given in the last raw.

Property	Young’s Modulus, E, GPa	Proof Stress, R_p0.2_, MPa	Tensile Strength R_m_, MPa	Elongation, e, %	Reduction of Area, A, %	Number of Samples
ASTM F2914 Requirements		825 min	895 min	10 min	15 min	
Average and standard deviation, As-built	118.8 ± 3.8	1036 ± 17	1122 ± 22	9.8 ± 3.8	11.5 ± 5.6	12
Average and standard deviation, HIP	119.5 ± 5.3	971 ± 20	1086 ± 19	18.9 ± 3.1	38.3 ± 11	6
Average and standard deviation, HIP	120.0 ± 5.8	976 ± 19	1090 ± 18	20.1 ± 0.7	42.8 ± 2.2	5

**Table 3 materials-12-01470-t003:** Averages and standard deviations, S, of the dynamic elastic moduli of as-built samples A3, H3, and O3. The columns indicate density, longitudinal velocity, V_L_, shear velocity, V_S_, Young’s modulus, E, shear modulus, G, and Poisson’s ratio, ν.

Sample Designation	Density [kg/m^3^]	V_L_ [m/s]	V_S_ [m/s]	E [GPa]	G [GPa]	ν
A3	4422 ± 3	6201 ± 1	3213 ± 2	120.2 ± 0.3	45.7 ± 0.1	0.316 ± 0.005
H3	4422 ± 3	6233 ± 4	3222 ± 2	121.0 ± 0.3	45.9 ± 0.1	0.318 ± 0.006
O3	4222 ± 3	6212 ± 14	3211 ± 2	120.1 ± 0.4	45.6 ± 0.1	0.318 ± 0.007

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
