# Peer review of "Mapping the Tray of Electron Beam Melting of Ti-6Al-4V: Properties and Microstructure"

_materials, 2019, doi:10.3390/ma12091470_

Reviewer 1 Report

The manuscript ‘Mapping the tray of electron beam melting of Ti-6Al-4V: Properties and microstructure’ by E. Tiferet et al. deals with 3D printing of columnar structures on a plate by EBM, their investigation function of their position on the printing board (and consequently their heating/cooling rate ) and after hot isostatic pressing.

In my opinion, this manuscript is interesting. During its lecture some questions rose and I think it would be beneficial that they be answered in the body of the manuscript to increase the clarity of the paper.

1.       What is a real application of HIP after 3D printing? Is there a way to produce shapes in such a manner? Why not HIP directly for powders? Why not casted samples+HIP?

2.       What is the reason to use powder heated at 550 and 700 C?

3.       What is the vacuum level during processing?

4.       Please define all terms in the manuscript even if they are widely known. Some examples are HR SEM, Leco

5.       The sections related to density variation of samples could be eliminated. The variations are too small and the errors could make the variations statistically irrelevant. A few explanations would be enough instead of Figs. 3 and 4

6.       Fig. 6- can you include the result of the measurement A14?

7.       Is it possible that the front and rear built borders to have a different microstructure than the rest of the samples, i.e. a martensitic structure as compared to an alpha + beta structure? This could explain the inferior mechanical properties.

8.       It is not known for the casual reader what Arcam 2X means. Please change in the first phrase of Abstract with “EBM printing machine” or a similar terminology

Author Response

1.       What is a real application of HIP after 3D printing? Is there a way to produce shapes in such a manner? Why not HIP directly for powders? Why not casted samples + HIP?

Answer 1: AM offers intricate geometries that will be designed for a specific demand. Residual flaws like lack of fusion, cracks or spherical porosity affect fatigue life. In order to reduce such flaws population post processing HIP significantly improves stress endurance, tensile elongation and reduce the mechanical properties variation.  Direct HIP for powders or even cast samples is a well-known technology [7], yet it offers limited shape options when compared to AM. HIP of cast sample is the most used in aerospace technology [7], yet casting of intricate shapes is even more complex than direct HIP.

Added to the introduction.  

2.       What is the reason to use powder heated at 550 and 700 C?

Answer 2: Main reason for conducting the preheating stage is to improve the energy deposition efficiency. When an electron beam interacts with a cold powder, part of the energy is not transferred into heat as intended. Instead, some powder gets charged by the electrons [[i]] and a kinematic phenomenon of recoiled powder particles emerging from the powder bed accrues. The preheating ensures the efficiency of the energy deposition is optimize.

Explanation added to section 2.3.   

3.       What is the vacuum level during processing?

Answer 3: Vacuum conditions were 10-5 mbar of initial vacuum, with a needle valve providing constant bleed of Helium to get an environment at a final pressure of 10-3 mbar.

Added to 2.3 section (Printing Theme).  

4.       Please define all terms in the manuscript even if they are widely known. Some examples are HR SEM, Leco

Answer 4: Done.

5.       The sections related to density variation of samples could be eliminated. The variations are too small and the errors could make the variations statistically irrelevant. A few explanations would be enough instead of Figs. 3 and 4.

Answer 5: Figure 4 was erased and text was re-edited.

6.       Fig. 6- can you include the result of the measurement A14?

Answer 6: It was included.

7.       Is it possible that the front and rear built borders to have a different microstructure than the rest of the samples, i.e. a martensitic structure as compared to an alpha + beta structure? This could explain the inferior mechanical properties.

Answer 7: The microstructure is similar. Our interpretation of the fractography (figure 11 in the paper) is that the source of the inferior properties emerges from larger amount and size of micro imperfections in the samples at the edges as compare to the amount in central raw. These imperfections were cured by the post HIP treatment.

 8.       It is not known for the casual reader what Arcam 2X means. Please change in the first phrase of Abstract with “EBM printing machine” or a similar terminology

Answer 8: Done

Many thanks for the comments

[i] Z. C. Cordero, H. M. Meyer, P. Nandwana, R. R. Dehoff, " Powder bed charging during electron-beam additive manufacturing", Acta Materialia 124 (2017) 437-445.

Reviewer 2 Report

This work presents the benefits of hot isostatic pressing for additive manufactured Ti-6Al-4V mechanical properties and microstructure. The work was well prepared, the experiments were designed properly, and the results were explored and explained sufficiently.

It is recommended to be published after several minor corrections.

A few minor units and equations were missing at line 133, 152-154 and some forms. Please review the manuscript carefully and correct them.

At line 116, the STL method was mentioned for the first time, please indicate the full name.

Author Response

A few minor units and equations were missing at line 133, 152-154 and some forms. Please review the manuscript carefully and correct them.

Done

At line 116, the STL method was mentioned for the first time, please indicate the full name.

Done

Reviewer 3 Report

Dear authors,

thank you for your sound and clear-written contribution to materials. I have only minor comments on your manuscript:

- line 151: if your standard deviation of density is in the range of 1.2kg/m^3 for a density of approx. 4430 kg/m^3, this is equal to 0.0027% relative density. You are discussion changes of relative density from 0.9991 to 0.9998 - this is way smaller than the standard deviation of your measurement. Are you sure you can really measure such differences?

Fig. 6b: y-axis title is shifted
Fig. 10: give meaningful titles to the axis labels, not only the symbol

I was not able to read the greek letters - did you use an unusual font? Please check before re-submitting.

Author Response

Thank you for your sound and clear-written contribution to materials. I have only minor comments on your manuscript:

- line 151: if your standard deviation of density is in the range of 1.2kg/m^3 for a density of approx. 4430 kg/m^3, this is equal to 0.0027% relative density. You are discussion changes of relative density from 0.9991 to 0.9998 - this is way smaller than the standard deviation of your measurement. Are you sure you can really measure such differences?

The uncertainty is 0.027% and we are sure. May we please refer you to our work published as: T.Sol,S.Hayun, D.Noiman ,E.Tiferet, O.Yeheskel, O.Tevet, “Nondestructive ultrasonic evaluation of additively manufactured AlSi10Mg samples” Additive Manufacturing, Volume 22, August 2018, Pages 700-707. Specifically, Appendix A. provides the SOP for these measurements

We wish to thank the reviewer for the comments

Round  2

Reviewer 1 Report

In my opinion the new version of the manuscript is clearer for the readers. I recommend the publication of the manuscript in its current form.